## [Transparent Peer Review file · Nature Communications]

Expanding the DNA Damaging Potential of Artificial Metallo-Nucleases with Click Chemistry

Corresponding Author: Professor Andrew Kellett

Version 0:

Reviewer comments:

Reviewer #1

(Remarks to the Author)

Gibney et al. present an elegant synthesis and characterization of a class of artificial metallonucleases. The additional characterization of TC-Py, which is a novel compound, has substantial evidence for the compounds activity, its DNA damaging potential, and its in vivo activity. The study is comprehensive and rigorous. There are few questions which should be addressed to ensure the interpretations are sound and the inferred potential activity is complete.

1) The authors do not comment in the discussion about why the TC-thio compound shows higher activity in the NCI-60 panel. While there is a separate paper for the TC-thio compound, the activity without copper is higher. Is this an advantage or disadvantage in the authors opinion? Why would TC-Py be a more attractive AMN target for cancer cells?

2) A recent study quantified the labile copper levels within the NCI-60 cell panel (PMID:39813247) contextualizing the results with the evidence from this paper might offer insight into the free molecules ability to leverage intracellular Cu for its activity.

3) What was the rationale for not repeating the cytotoxicity with the copper containing TC-Py and TC-Thio with the MDA-MB-231 cells used for ICP-MS? Why was DNA damage testing not performed in these cell lines as well to confirm mechanisms?

4) The ICP-MS uses MDA-MB-231 cells, but the import into the nucleus might differ for cell lines with high or low levels of copper transporters. Were other cell lines examined? Nuclear penetrance may be important for mechanism of action. Unless the authors expect the Py molecule to interact and interfere with RNA processing. Do they expect RNA or DNA levels reductions in human cells treated with the free or Cu-bound compound? The bacterial evidence doesn't sufficiently address the potential nuclear DNA damage induction issues with low nuclear penetrance across cell models. The PBMC work is interesting, but it is unclear what the relative levels of DNA damage induction are compared to other models. Also, the nuclear uptake in the PBMCs is unknown. Since cell survival isn't assessed it is a gap in the manuscript. The GI50s range indicates additional cellular mechanisms may impact the AMN activity.

5) For the RADD assay, the authors use EndoIV and APEI which have similar activities, what is the rationale for adding both enzymes?

6) The RecA-GFP data show no formation of RecA-GFP foci at 10 or 30 min of treatment compared to the control.

Additionally, the DAPI staining is less intense. Could this be an effect of DNA binding by TC-Py that precludes DAPI staining and may hinder recruitment? The authors have not discussed this sufficiently. The compaction changes are similar to nitrofurantoin which does form foci. The pulsed-field gel only shows the highest dose and 30 min time point. The combing shows shortening, but the other results show some mechanistic inconsistencies.

7) Other minor issues: Cell media and other relevant details for the MDA-MB-231 are also not provided. The cell number and days of growth prior to ICP-MS analysis should be specified. Also in the methods "Salinization of coverslips"- should this be silanization?

Reviewer #2

(Remarks to the Author)

Reviewing report

Title " Expanding the DNA Damaging Potential of Artificial Metallo-Nucleases with Click Chemistry"

Authors: Andrew Kellett et al.

(Manuscript Number: NCOMMS-25-24289)

For the current investigation, the authors used a click approach to prepare some Tri-Click ligands and evaluated their

artificial metallo-nucleases (AMN) activities. Among these compounds, the copper complex of Tri-Click-Pyridine (Cu₃-TC-Py) was found to display significant potential. They anticipated the emergence of Cu₃-TC-Py as a lead AMN with high reactivity for DNA damage applications central to anticancer therapy.

However, this manuscript faces some major and minor concerns that should be addressed before being considered for publication in Nature Communications as follows:

- In several parts of the manuscript, the authors used the expression "library of compounds" for just six products, which is weird for me. The variety of ligands structures is still not high.
- For the graphical abstract: the authors wrote the "click chemistry" beneath the clicked product, not for the process (reaction itself).
- The authors used ESMS for characterisation of the Cu(II) binding properties of TC ligands, I am wondering if there is not additional tool or techniques that could support this binding?
- For reference section: the authors did not take sufficient time to write the references in a proper form; for example, they used (one author et al.) in most references, which is not accepted in high-ranked journals. Moreover, they used the full names for given names in some references, such as in ref 32. (Zuin Fantoni), and ref. 36 (Kamal El-Sagheir).
- Too much assumed knowledge in the captions of figures that reach 14 lines in some cases, as in Fig. 4, this could be shortened to help readability.
- M⁺ and M⁺²: the positive charge should be superscripted in the whole manuscript and also in the supporting information file. Similarly, ¹H & ¹³C NMR and other abbreviations.
- In some cases, the authors used a mixture of (full names + abbreviations) for some expressions such as water:DMF.
- Figure 1: insert the reference for a) previous work.
- In figure 1d: Qhoechst "h" of hoechst is should be capitalized.
- An in-depth analysis of MD simulation outcomes should be done. Regarding the molecular dynamics (MD) simulation study, authors should properly explain the implications of the parameters, i.e., RMSD, RMSF, and Rg, etc., in context with simulated complex stability.
- In supplementary file:
 - a) line 451 Figure S9: ¹³C NMR of TC-Amide in DMSO-d₆: I think this spectrum belongs to "TC-Pyrm".
 - b) Charts (ESI-MS, ¹H, ¹³C NMR spectra) of TC-Py are missed. Moreover, the chart of ¹³C NMR spectrum of TC-Benzo is also missing.
 - c) The authors have four digits for the ESI-MS, but they used just two digits in writing
- For language: There are a number of misspellings and grammar errors, so the authors should revise and correct them. The punctuation should be revised carefully. Just examples: Line 85 sample should be changed to "samples", TC-ACID is written in other cases TC-Acid, etc....

Prof. Dr. Sherif Shaban Ragab
(<https://orcid.org/0000-0003-0474-4755>)

Reviewer #3

(Remarks to the Author)

The authors propose a Tri-Click chemistry approach for developing artificial metallo-nucleases (AMNs) having DNA-damaging properties. They have synthesised a library of six new Tri-Click ligands incorporating systematic cyclic and acyclic hetero atom donors capable of forming coordination bonds with transition metals. The three triazoles generated by Cu-catalysed alkyne-azide cycloaddition have been characterised for their ability to coordinate with copper, and some of the ligands show coordination with three copper ions as well as binding ability with DNA. Among these, Tri-Click-Pyridine (Cu₃-TC-Py) and Tri-Click-Pyrimidine (Cu₃-TC-Pyrm) were found to have good binding affinity. Based on the ability to form a coordination bond and DNA binding affinity, Cu₃-TC-Py were further utilised for in-cell penetration, bacterial DNA damage repair and in silico experiments. Although the in vitro experiments data are solid, in vivo experiments regarding the DNA damage in cells are very minimal.

Key points:

- 1) It is currently unclear whether Cu₃-TC-Py exhibits selectivity toward a specific DNA sequence or structural motif. If the compound binds non-specifically to any DNA, its therapeutic application may be limited due to potential off-target effects on normal cells. Investigating whether Cu₃-TC-Py preferentially binds to specific DNA mismatches or specific DNA conformations would be valuable.
 - 2) Experimental structural validation of the DNA–Cu₃-TC-Py complex would significantly strengthen the study. Given the relevance of artificial metallo-nucleases in DNA recognition, structural elucidation via NMR spectroscopy or X-ray crystallography would provide more definitive insight into the binding mode and enhance the impact of the findings.
 - 3) In the biophysical experiments, the authors suggest that Cu₃-TC-Py occupies a binding site of 6 base pairs to DNA. However, the molecular docking and simulation data do not clearly illustrate whether the compound indeed spans 6 base pairs within the dodecamer sequence. The authors should clarify whether the computational findings are consistent with the experimental observations.
 - 4) Given the authors previous publications detailing Tri-Click-ligands as AMNs (Nucleic Acids Research, 49, 18, 2021, 10289–10308; Angew. Chem. Int. Ed. 2023, 62, e202305759), the current manuscript's expansion of AMN ligands via the same tri-click strategy appears to lack sufficient novelty for a new contribution.
- In conclusion, the paper presents an innovative click chemistry-based strategy for developing potential anticancer agents.

The approach is scientifically sound, and the focus on oxidative DNA damage is promising. However, expanding on the above-mentioned aspects of specificity and binding mechanism would significantly strengthen the manuscript. Given these points, the current manuscript seems more suitable for a specialised journal.

Minor comments:

1. The Figure 4 legends: panels a and c need to be corrected.
2. Information about the presence of multiple-testing corrections is missing from the statistical test details.
3. To improve the readability of the main text, please use the full name for ligand side chains before using the abbreviation

Version 1:

Reviewer comments:

Reviewer #1

(Remarks to the Author)

The authors have done a terrific job addressing the comments. The additional experiments also added to the significance of the work and clarify mechanisms of action.

Reviewer #2

(Remarks to the Author)

The authors have satisfactorily addressed all previous comments, and the revised manuscript shows clear improvement in quality and presentation. It now meets the publication standards of the journal and is recommended for acceptance.

Reviewer #3

(Remarks to the Author)

The revised manuscript and the response letter have been reviewed. The authors have effectively addressed all concerns and suggestions, resulting in a significantly improved study. Specifically, the clarifications and the inclusion of new data particularly the FRET melting assays with diverse hairpin sequences to confirm Cu³⁺-TC-Py binding specificity and the cellular Cu internalization studies are noteworthy. In summary, the authors have demonstrated good work, and I recommend revised manuscript for acceptance.

NCOMMS-25-24289

At the outset, we thank all reviewers for their detailed and constructive comments, valuable suggestions, and for recommending our manuscript for publication in *Nature Communications*. We have carefully addressed all points raised and, where appropriate, performed additional experiments to further support our conclusions.

Reviewer #1 (Remarks to the Author):

Gibney et al. present an elegant synthesis and characterization of a class of artificial metallonucleases. The additional characterization of TC-Py, which is a novel compound, has substantial evidence for the compounds activity, its DNA damaging potential, and its in vivo activity. The study is comprehensive and rigorous. There are few questions which should be addressed to ensure the interpretations are sounds and the inferred potential activity is complete.

Q1) The authors do not comment in the discussion about why the TC-thio compound shows higher activity in the NCI-60 panel. While there is a separate paper for the TC-thio compound, the activity without copper is higher. Is this an advantage or disadvantage in the authors opinion? Why would TC-Py be a more attractive AMN target for cancer cells?

A1) We thank the reviewer for this comment. We believe that TC-Py is a more promising agent as it arose from the structural variation screen presented in the early part of this manuscript. This ligand showed an overall higher DNA binding affinity, more potent DNA damaging properties, and is likely a more stable complex than the earlier reported Cu(II)-TC-Thio, given the direct differences we observed in ESI-MS spectra. To help support this finding, additional UV-vis absorbance experiments have been performed and are now discussed in the “*Copper binding analysis*” section. Furthermore, additional cytotoxicity experiments have been performed comparing Cu₃-TC-Py and Cu₃-TC-Thio, introduced in the “*Cytotoxicity and Cu Internalisation Studies*” section of the paper (*vide infra*), that demonstrates superior anticancer potential by the copper(II) bound TC-Py complex. We would also like to highlight that earlier DFT calculations with TC-Thio demonstrated that Cu-S coordination of TC-Thio was only possible in the reduced, Cu(I), state. While Cu(I) predominates in the intracellular environment, the complex cycles between Cu(I) and Cu(II) when mediating ROS activity. Therefore, we believe a persistent bidentate binding mode (as would be the case with Cu₃-TC-Py) could present an advantage in this complex series. This point is now discussed in the conclusion, lines 473-477.

Q2) A recent study quantified the labile copper levels within the NCI-60 cell panel (PMID:39813247) contextualizing the results with the evidence from this paper might offer insight into the free molecules ability to leverage intracellular Cu for its activity.

A2) We thank the reviewer for highlighting this important work. We have now revised the manuscript to contextualise GI₅₀ values with the previously reported intracellular Cu levels. In lines 276-286 of our revision, we included the following text:

“Notably, growth inhibition by the free ligands does not appear to correlate directly with either labile or total intracellular Cu levels within the NCI-60 panel, as carefully characterised by Chang et al.³² For instance, although PC3, BT-549, and HOP-92 cells exhibit some of the highest labile intracellular Cu levels, exposure to the TC-Py ligand resulted in low micromolar

GI50 values (14.45, 12.59, and 1.20 μM , respectively), comparable to those observed in cell lines with relatively low labile Cu, such as MCF7, T47D, and OVCAR-8 (9.12, 6.03, and 2.69 μM , respectively). While additional studies would be required to clarify the underlying processes, these findings suggest that a simple prodrug mechanism—whereby the ligands chelate intracellular Cu and elicit cytotoxicity through subsequent DNA damage—does not appear to be the dominant pathway responsible for growth inhibition by the free TC ligands.”

Q3) What was the rationale for not repeating the cytotoxicity with the copper containing TC-Py and TC-Thio with the MDA-MB-231 cells used for ICP-MS? Why was DNA damage testing not performed in these cell lines as well to confirm mechanisms?

A3) We thank the reviewer for highlighting this limitation. We have now broadened the study to include a direct cytotoxicity screen using $\text{Cu}_3\text{-TC-Py}$ and $\text{Cu}_3\text{-TC-Thio}$ on MDA-MB-231 and other cell lines with varying intracellular Cu profiles including PC3, DU145, and A549. The results are now presented in **Figure 3b** and discussed in lines 287-298 as follows:

“To further examine this effect, we evaluated the sensitivity of several cell lines with distinct intracellular Cu lability profiles to $\text{Cu}_3\text{-TC-Py}$ and $\text{Cu}_3\text{-TC-Thio}$, alongside CuCl_2 as a control (Figure 3b and S29). In cell-based assays, CuCl_2 served as the copper source, and CuCl_2 -only controls were tested at three-times the molar concentration of the $\text{Cu}_3\text{-TC}$ complexes to ensure equivalent total Cu exposure. We selected four cell lines—PC3, DU145, A549 and MDA-MB-231—for cytotoxicity assays. PC3 exhibits a median level of total intracellular Cu, but among the highest levels of labile intracellular Cu whereas DU145 displays the opposite profile, with high total, but low labile, levels of Cu. Both A549 and MDA-MB-231 contain relatively low levels of both Cu pools. In these assays, CuCl_2 alone showed no activity up to 250 μM , whereas $\text{Cu}_3\text{-TC-Py}$ consistently produced IC_{50} values in the 10-30 μM range and $\text{Cu}_3\text{-TC-Thio}$ in the 30-60 μM range. Notably, $\text{Cu}_3\text{-TC-Thio}$ displayed limited toxicity toward DU145 cells within the tested range, while $\text{Cu}_3\text{-TC-Py}$ yielded an IC_{50} of 12 μM for the same cell line. Collectively, these data indicate that $\text{Cu}_3\text{-TC-Py}$ exhibits greater anticancer potency than $\text{Cu}_3\text{-TC-Thio}$ across this panel.”

Q4) The ICP-MS uses MDA-MB-231 cells, but the import into the nucleus might differ for cell lines with high or low levels of copper transporters. Were other cell lines examined? Nuclear penetrance may be important for mechanism of action. Unless the authors expect the Py molecule to interact and interfere with RNA processing. Do they expect RNA or DNA levels reductions in human cells treated with the free or cu-bound compound? The bacterial evidence doesn't sufficiently address the potential nuclear DNA damage induction issues with low nuclear penetrance across cell models. The PBMC work is interesting, but it is unclear what the relative levels of DNA damage induction are compared to other models. Also, the nuclear uptake in the PBMCs is unknown. Since cell survival isn't assessed it is a gap in the manuscript. The GI50s range indicates additional cellular mechanisms may impact the AMN activity.

A4) We thank the reviewer for their comment and for highlighting this important point. To investigate the relationship between intracellular copper levels and cytotoxicity, we repeated total cellular Cu uptake assays in PC3, MDA-MB-231, and A549 cells using both $\text{Cu}_3\text{-TC-Py}$ and $\text{Cu}_3\text{-TC-Thio}$ complexes. To maintain consistency with the subsequent biological experiments and functional assays reported in this work, the new ICP-MS measurements used copper(II) chloride as the copper source, rather than copper(II) nitrate as in the original

submission. The updated results are shown in **Figure 4a** and discussed in lines 299–306, as quoted below:

“Next, to understand the Cu uptake properties of Cu₃-TC-Py, we conducted ICP-MS experiments to measure the total Cu content in A549, MDA-MB-231, and PC3 cell lines post-treatment with Cu₃-TC-Py at 48 and 72 h (Figure 4a). The results show that Cu₃-TC-Py significantly increased the Cu content in all cell lines at both time points, suggesting significant cellular uptake of the complex. For MDA-MB-231 and PC3 cells, a similar uptake pattern emerged where ~20 ng of Cu / million cells was found post 48 h of exposure with this level rising to ~40 ng after 72 h of exposure. A different trend emerged for A549 cells where very high Cu levels (~50 ng of Cu / per million cells) were detected after 48 h with lower levels (~30 ng) detected over 72 h, suggesting a degree of Cu efflux upon prolonged exposure. A different trend emerged for A549 cells where very high Cu levels (~50 ng of Cu / per million cells) was detected after 48 h of exposure with lower levels (~30 ng) detected after 72 h, suggesting a degree of Cu efflux.”

We next conducted cell-cycle analysis, which revealed a marked accumulation of cells in the G₂/M phase with corresponding decreases in G₀/1 and S populations. This profile is characteristic of a DNA-damage response pathway that contributes to the overall cytotoxic mechanism of the complex. This data is highlighted in the main manuscript in Figure 4c and in lines 308–313, quoted below:

“Finally, cell cycle analysis of MDA-MB-231 cells revealed that treatment with 25 μM Cu₃-TC-Py resulted in an increased population within the G₂/M phase and a corresponding decrease in S and G₀/1 phases, relative to cells treated with CuCl₂ alone at 75 μM (Figure 4b). This distribution is consistent with activation of the G₂/M DNA damage checkpoint, which prevents mitosis until DNA is repaired.^{33–35} In summary, these results indicate that Cu₃-TC-Py is cytotoxic and likely acts predominantly via the induction of DNA damage.”

Regarding the comment “Since cell survival isn’t assessed..” we would like to highlight that an additional cytotoxicity screen was conducted (presented in Figure 3b) as discussed earlier. Finally, we agree it is plausible that other cellular mechanisms are responsible for the inhibitory effects of the TC-Py ligand alone. As such, we included a new line 281-284 (discussed earlier): *“While additional studies would be required to clarify the underlying processes, these findings suggest that a simple prodrug mechanism—whereby the ligands chelate intracellular Cu and elicit cytotoxicity through subsequent DNA damage—does not appear to be the dominant pathway responsible for growth inhibition by the free TC ligands.”*

Q5) For the RADD assay, the authors use EndoIV and APEI which have similar activities, what is the rationale for adding both enzymes?

A5) We thank the reviewer for highlighting the need for clarity here. APE I and ENDO IV do indeed have overlapping substrate profiles in that both cleave to the 5' end of AP sites. ENDO IV, however, has a broader substrate scope, with processing of 3' blocking groups like 3'-α, β-unsaturated aldehyde, phosphoglycoaldehyde. Using both enzymes ensured a more comprehensive capture of total DNA oxidation. To clarify this in the main text we have added the following to lines 380-385.

“Both Endo IV and APE I were employed to maximise recognition of abasic and oxidised abasic lesions. APE I preferentially cleaves canonical apurinic/aprimidinic sites within duplex

DNA, whereas Endo IV also recognises oxidised and structurally distorted abasic sites and 3' blocking groups. Their combined use therefore broadens detection sensitivity toward the diverse oxidative lesions generated by copper redox cycling”

Q6) The RecA-GFP data show no formation of RecA-GFP foci at 10 or 30 min of treatment compared to the control. Additionally, the DAPI staining is less intense. Could this be an effect of DNA binding by TC-Py that precludes DAPI staining and may hinder recruitment? The authors have not discussed this sufficiently. The compaction changes are similar to nitrofurantoin which does form foci. The pulsefield gel only shows the highest dose and 30 min time point. The combing shows shortening, but the other results show some mechanistic inconsistencies.

A6) We thank the reviewer for this helpful comment. We have clarified in the revised manuscript that the apparent absence of RecA foci is best explained by rapid nucleoid degradation, as Cu₃-TC-Py disperses the nucleoid within 10 min. This is more rapid than nitrofurantoin, which only shows RecA foci at earlier timepoints. We also performed a new DnaN localisation assay, which confirmed that protein delocalization results from nucleoid fragmentation rather than competitive DAPI binding. These explanations and the supporting data are now presented in the updated *Cytological Profiling* section (lines 396-445) and associated figures (Figures 4g–h, Figures S32–S37). The updated text now reads as follows:

*“To probe the structural changes imposed on DNA by Cu₃-TC-Py, the AMN activity was probed in bacterial cells using a combination of cytological profiling, using phase contrast microscopy and DAPI DNA staining, functional profiling using GFP-tagged RecA (which is an essential protein for maintaining and repairing DNA in bacteria) and DnaN (the beta subunit of DNA polymerase III), and single-molecule analysis. We first treated *Bacillus subtilis* with Cu₃-TC-Py for 10 and 30 min before staining and imaging via phase contrast and fluorescence microscopy (**Figure 4g**). DNA targeted agents can be expected to cause a change (compaction or relaxation) of the bacterial nucleoid while a DNA damaging agent such as an AMN may be expected to increase recruitment of RecA and trigger a resultant increase in RecA foci. Ciprofloxacin and nitrofurantoin were used as controls in these experiments. Ciprofloxacin inhibits DNA gyrase and topoisomerase IV causing defects in DNA replication and nucleoid separation resulting in clear nucleoid compaction and recruitment of the RecA protein to single-stranded DNA arising from strand breaks. Nitrofurantoin is a prodrug that is activated by cellular nitroreductases leading to the formation of reactive species that damage cellular macromolecules, most prominently DNA, causing nucleoid relaxation and, at high doses, destruction of the entire nucleoid. Relative to the positive controls treated with ciprofloxacin and nitrofurantoin, Cu₃-TC-Py resulted in an apparent loss of DAPI staining together with limited recruitment of GFP-RecA foci (**Figure S32 and S33**). Given the earlier evidence of combined AMN and DNA condensation activity by Cu₃-TC-Py, we hypothesised this cytological profile may arise due to near-total degradation of the genetic material. Therefore, we conducted image analysis to identify the DNA compaction ratio within imaged cells (**Figure 4h**) where the compaction ratio is an expression of the nucleoid volume relative to the total cell volume. Theoretically, DNA degradation causes a decrease in the DNA compaction ratio as the nucleoid is dispersed, while compaction would have the inverse effect. Data here showed that after 30 min, Cu₃-TC-Py significantly decreased the DNA compaction ratio, producing a similar profile to nitrofurantoin, thus demonstrating Cu₃-TC-Py damages and disperses the genetic material. It should be noted that nitrofurantoin followed a slower kinetic*

profile, characterized by RecA foci appearing at 10 min, followed by nucleoid relaxation, loss of DAPI signal, and loss of RecA foci at 30 min, while Cu₃-TC-Pyr already showed DNA dispersal at 10 min (**Figure S34**). This faster effect explains the absence of a high number of RecA foci in Cu₃-TC-Pyr treated samples as RecA cannot form filaments when the nucleoid is physically disintegrated. Since Cu₃-TC-Pyr showed preferential binding to the minor groove, which is also the binding site of DAPI, the lower signal intensity could be attributed to competition for this binding site. However, competitive binding would not explain the absence of RecA foci as the localization of DNA-binding proteins is not affected by DAPI staining. Indeed, the DNA-binding DNA polymerase III subunit, DnaN, localizes in the expected nucleoid-associated foci in untreated, but DAPI-stained, cells, but is dispersed after treatment with Cu₃-TC-Py (**Figures S35 and S36**).⁴¹ Since a loss of DAPI signal is also observed with nitrofurantoin, which does not bind to DNA but disperses the nucleoid in a similar manner as Cu₃-TC-Py, nucleoid fragmentation is a considerably more likely explanation for the dispersed DAPI signal and the loss of protein localization than competition between DAPI and Cu₃-TC-Py for the minor groove, although it cannot be excluded that the latter may partially contribute to the reduced DAPI signal.

Independent confirmation of this nucleoid-degrading mechanism was then sought using a combination of gel electrophoresis and single-molecule analysis, choosing the 30 min timepoint at which both Cu₃-TC-Pyr and nitrofurantoin showed nucleoid dispersal. Gel electrophoresis experiments involved treating cells in an identical manner to those used for the image analysis presented in **Figure 4f**. Thereafter, the total DNA content was extracted and visualised using pulse-field agarose gel electrophoresis, where changes in DNA molecule sizes are clearly identifiable. Untreated samples contained relatively uniform DNA molecules while those treated with Cu₃-TC-Py demonstrated reduced overall DNA content together with fragmentation patterns indicative of DNA ablation (**Figure S37**). Tandem single-molecule analysis experiments were then performed using the same treatment and extraction steps. Here, DNA was stained using YOYO-1, stretched on cover slides and imaged (**Figure 4i**). Shortening of DNA molecules was evident as untreated samples contained high counts of molecules in the 40-80 μm range, while samples treated with Cu₃-TC-Py and nitrofurantoin contained limited numbers of molecules above 40 μm and a significantly increased density of molecules below 20 μm.”

Q7) Other minor issues: Cell media and other relevant details for the MDA-MB-231 are also not provided. The cell number and days of growth prior to ICP-MS analysis should be specified. Also in the methods “Salinization of coverslips”- should this be silanization?

A7) We thank the reviewer for highlighting these errors. The revised materials and methods section now includes this detail. “Salinization” has also been corrected to “Silanization”.

Reviewer #2 (Remarks to the Author):

Reviewing report

Title " Expanding the DNA Damaging Potential of Artificial Metallo-Nucleases with Click Chemistry"

Authors: Andrew Kellett et al.

(Manuscript Number: NCOMMS-25-24289

For the current investigation, the authors used a click approach to prepare some Tri-Click ligands and evaluated their artificial metallo-nucleases (AMN) activities. Among these

compounds, the copper complex of Tri-Click-Pyridine (Cu₃-TC-Py) was found to display significant potential. They anticipated the emergence of Cu₃-TC-Py as a lead AMN with high reactivity for DNA damage applications central to anticancer therapy.

However, this manuscript faces some major and minor concerns that should be addressed before being considered for publication in Nature Communications as follows:

Q8) In several parts of the manuscript, the authors used the expression "library of compounds" for just six products, which is weird for me. The variety of ligands structures is still not high.

A8) We thank the reviewer for highlighting this and we have replaced mentions of "Library" with "Series" as this more accurately portrays the number and variation of compounds presented.

Q9) For the graphical abstract: the authors wrote the "click chemistry" beneath the clicked product, not for the process (reaction itself).

A9) This has now been corrected.

Q10) The authors used ESMS for characterisation of the Cu(II) binding properties of TC ligands, I am wondering if there is not additional tool or techniques that could support this binding?

A10) We thank the reviewer for highlighting this. Although X-ray crystallography would have been the ideal technique to confirm the binding of TC-Py with copper, our attempts using multiple crystallisation screens with various solvents, loading ratios, and techniques did not yield high-resolution structures. Therefore, we conducted a UV-vis continuous variation titration using TC-Py and CuCl₂. These data are presented in the revised supplementary information and were introduced in the main manuscript as follows (lines 90-95):

"To corroborate these findings, a UV-vis analysis was performed by adding Cu(NO₃)₂ to a fixed, 100 μM solution of TC-Py and monitoring the change in absorbance between 270 and 800 nm (Figure S20). Plotting of absorbance at 283 nm vs stoichiometric equivalents of Cu(NO₃)₂ showed a clear hyperchromic trend with an inflection point calculated at 2.93 equivalents of Cu(NO₃)₂, further supporting the formation of the trinuclear complex, Cu₃-TC-Py."

Q11) For reference section: the authors did not take sufficient time to write the references in a proper form; for example, they used (one author et al.) in most references, which is not accepted in high-ranked journals. Moreover, they used the full names for given names in some references, such as in ref 32. (Zuin Fantoni), and ref. 36 (Kamal El-Sagheir).

A11) We thank the reviewer for highlighting this. We have now used the correct citation style for Nature Communications and will confirm this style is appropriate with the editor.

Q12) Too much assumed knowledge in the captions of figures that reach 14 lines in some cases, as in Fig. 4, this could be shortened to help readability.

A12) We have now shortened the captions for figures 2 and 4, while retaining sufficient detail to aid with graphical interpretation.

Q13) M⁺ and M⁺²: the positive charge should be superscripted in the whole manuscript and also in the supporting information file. Similarly, ¹H & ¹³C NMR and other abbreviations.

A13) We have made the indicated changes throughout the main manuscript and the supporting information (NMR figure captions and Table S2).

Q14) In some cases, the authors used a mixture of (full names + abbreviations) for some expressions such as water:DMF.

A14) We have changed the DMF abbreviation to dimethyl formamide and checked the document for similar abbreviations.

Q15) Figure 1: insert the reference for a) previous work.

A15) We have added the requested citations.

Q16) In figure 1d: Q_{hoechst} "h" of hoechst is should be capitalized.

A16) We have now changed Q_{hoechst} to Q_{Hoechst} throughout the manuscript.

Q17) An in-depth analysis of MD simulation outcomes should be done. Regarding the molecular dynamics (MD) simulation study, authors should properly explain the implications of the parameters, i.e., RMSD, RMSF, and R_g, etc., in context with simulated complex stability.

A17) We thank the reviewer for their comment. We have added an expanded MD discussion in the supporting information (Figures S38-S41 and tables S4-S8), analysing RMSD, R_g and interaction types. We have updated the MD section of the main manuscript to summarise the additional findings as reproduced below (line 236-250).

"A detailed discussion of MD data is provided in the supporting information. Briefly, the plateauing of the Cu₃-TC-Py RMSD within ~2 μs indicates formation of well-equilibrated binding poses, with small fluctuations consistent with molecular conformational shifts and dynamics in the solvated DNA-bound structures. The radius of gyration (R_g) data shows a tendency for more pronounced and abrupt transition to more compact DNA structures in the minor groove-bound structure, supporting the hypothesis of complex-induced DNA compaction during the two-phase binding/condensation process observed experimentally (Figure S40). To probe the binding footprint, we quantified the frequency and character of base-specific interactions across all replicates, consistently revealing engagement with six base pairs from the minor groove (Tables S4-S8), which is in excellent agreement with our earlier FRET analysis with the F-DDH sequence."

Q18) Supplementary file:

a) line 451 Figure S9: ^{13}C NMR of TC-Amide in DMSO- d_6 : I think this spectrum belongs to "TC-Pyrm".

A18) We thank the reviewer for highlighting this error – we have corrected the figure caption.

Q19) Charts (ESI-MS, ^1H , ^{13}C NMR spectra) of TC-Py are missed. Moreover, the chart of ^{13}C NMR spectrum of TC-Benzo is also missing.

A19) We have included the ^1H , ^{13}C and ESI-MS spectra of TC-Py (Figures S8, S9 and S17) and the ^{13}C NMR of TC-Benzo (Figure S13).

Q20) The authors have four digits for the ESI-MS, but they used just two digits in writing

• For language: There are a number of misspellings and grammar errors, so the authors should revise and correct them. The punctuation should be revised carefully. Just examples: Line 85 sample should be changed to "samples", TC-ACID is written in other cases TC-Acid, etc....

A20) We thank the reviewer for highlighting this. We have reviewed the document carefully to improve consistency and readability.

Prof. Dr. Sherif Shaban Ragab

(<https://orcid.org/0000-0003-0474-4755>)

Reviewer #3 (Remarks to the Author):

The authors propose a Tri-Click chemistry approach for developing artificial metallo-nucleases (AMNs) having DNA-damaging properties. They have synthesised a library of six new Tri-Click ligands incorporating systematic cyclic and acyclic hetero atom donors capable of forming coordination bonds with transition metals. The three triazoles generated by Cu-catalysed alkyne-azide cycloaddition have been characterised for their ability to coordinate with copper, and some of the ligands show coordination with three copper ions as well as binding ability with DNA. Among these, Tri-Click-Pyridine (Cu₃-TC-Py) and Tri-Click-Pyrimidine (Cu₃-TC-Pyrm) were found to have good binding affinity. Based on the ability to form a coordination bond and DNA binding affinity, Cu₃-TC-Py were further utilised for in-cell penetration, bacterial DNA damage repair and in silico experiments. Although the in vitro experiments data are solid, in vivo experiments regarding the DNA damage in cells are very minimal.

Key points:

Q21) It is currently unclear whether Cu₃-TC-Py exhibits selectivity toward a specific DNA sequence or structural motif. If the compound binds non-specifically to any DNA, its therapeutic application may be limited due to potential off-target effects on normal cells. Investigating whether Cu₃-TC-Py preferentially binds to specific DNA mismatches or specific DNA conformations would be valuable.

A21) We thank the reviewer for highlighting this gap. We have now performed additional FRET melting assays using hairpins with varied sequence contexts and DNA base pair mismatches. More specifically, we used methyl cytosine and uracil bases in the DDH construct to investigate groove specificity, and we additionally varied the GC content to investigate sequence selectivity. We also introduced mismatches to probe for metalloinsertion interactions. This new data is presented in the revised **Figure 2C**, and introduced the text (lines 160-184), which reads:

“We next employed the FRET melting assay within a wider panel of hairpin sequences to identify preferential groove or sequence context binding by Cu₃-TC-Py. Here, we designed a number of DNA hairpins with varying groove accessibilities, GC content, along with base pair mismatches (Figure 2c and Figure S24). We first modified the DDH sequence to contain 5-methyl cytosine (MDDH) or uracil (UDDH) bases in place of cytosine and thymine, respectively. The methyl group of 5-methyl cytosine reduces steric accessibility to the major groove, while the absence of the methyl group in uracil (compared to thymine) increases the accessibility of the major groove. Cu₃-TC-Py induced thermal melting changes (ΔT_m) of +6.65 °C for MDDH and +3.00 °C for UDDH with corresponding K_d values of 2.61 and 0.68 μ M, respectively. Together these results indicate that Cu₃-TC-Py can bind both DNA grooves with high affinity, but minor groove residency results in greater thermodynamic stabilisation of the duplex.

Next, to investigate the sequence selectivity of Cu₃-TC-Py, we used hairpins with varying sequence contexts. The D6aH and TP hairpins provided targets with low and high GC content, respectively. Here, Cu₃-TC-Py produced a ΔT_m of +12.20 °C for the TP sequence, but had negligible impact on the melting temperature of D6AH, suggesting a high preference for GC-rich tracts. The TP sequence contained a single AT/AT step in the termini of the duplex, which some copper(II) compounds have shown selectivity towards.²⁶ To evaluate if the thermodynamic stabilisation of TP was due to specific binding within a contiguous GC-rich tract, experiments with the ATH hairpin, which contains a central AT/AT step flanked by GC rich regions, were performed. Although Cu₃-TC-Py significantly stabilised the ATH sequence, the ΔT_m value (+7.89 °C) was lower than TP. Finally, we designed DDH analogues containing single- (SMH) and double-mismatches (DMH) to investigate if Cu₃-TC-Py could bind via metalloinsertion—a binding mode distinctive to intercalation, that causes the ejection of mismatched bases induced by a portion of a bulky metal complex sitting directly within the DNA helix.²⁷ Here, we found that both the single- and double-mismatches completely negated the thermal stabilisation provided by Cu₃-TC-Py, ruling out metalloinsertion as a potential binding mode. Overall, these FRET-based thermal melting experiments suggest that Cu₃-TC-Py shows a significant degree of selectivity for the minor groove of GC-rich DNA, with limited tolerance for DNA mismatches or AT-rich tracts within the binding region.”

Q22) Experimental structural validation of the DNA–Cu₃-TC-Py complex would significantly strengthen the study. Given the relevance of artificial metallo-nucleases in DNA recognition, structural elucidation via NMR spectroscopy or X-ray crystallography would provide more definitive insight into the binding mode and enhance the impact of the findings.

A22) We thank the reviewer for this comment, and we agree that crystallographic data of the Cu₃-TC-Py complex bound to DNA would undoubtedly strengthen the manuscript. A similar question was raised by reviewer #2 (see A10). We would like to add that the the crystallisation

of DNA damaging complexes, particularly those with 'self-activating' properties such as this class, with oligonucleotides is especially difficult as the complex degrades the DNA sample. While we have had recent success crystallising Ru(II) polypyridyl complexes (<https://doi.org/10.1002/anie.202318863> and <https://doi.org/10.1039/D4SC01448K>), unfortunately our efforts to crystallise copper artificial metallo-nucleases with oligonucleotides have not been successful to date. Although we have shown data here for DNA cleavage in the presence of a reductant (Na-L-ascorbate) we have identified that Cu₃-TC-Py shows self-activation properties, whereby DNA is damaged in a short timeframe (30 min) in the absence of a reductant. Figure R1 below, included here for the purpose of fully addressing this question, demonstrates topoisomerase inhibition upon unwinding negatively supercoiled pUC19 plasmid DNA to the positive superhelical form. However, at higher Cu₃-TC-Py concentrations, the pUC19 DNA is clearly relaxed to the open circular form (highlighted in the red box), despite the lack of added reductant.

Figure R1: Agarose gel of topoisomerase inhibition assay, showing clear self-activated cleavage at high concentrations (red).

Q23) In the biophysical experiments, the authors suggest that Cu₃-TC-Py occupies a binding site of 6 base pairs to DNA. However, the molecular docking and simulation data do not clearly illustrate whether the compound indeed spans 6 base pairs within the dodecamer sequence. The authors should clarify whether the computational findings are consistent with the experimental observations.

A23) We thank the reviewer for highlighting this. We have, in response to this comment and related comments from Reviewer #2, included a significantly expanded discussion of MD data in the supporting information, which now includes analysis of the interaction mode and binding site size in the simulations (see A17). With specific regard to the binding site size, we have included the following in the main text from lines 242-250:

“to probe the binding footprint, we quantified the frequency and character of base-specific interactions across all replicates, consistently revealing engagement with six base pairs from the minor groove (Tables S5-S9), which is in excellent agreement with our earlier FRET analysis with the F-DDH sequence.”

Q24) Given the authors previous publications detailing Tri-Click-ligands as AMNs (Nucleic Acids Research, 49, 18, 2021, 10289–10308; Angew. Chem. Int. Ed. 2023, 62, e202305759), the current manuscript's expansion of AMN ligands via the same tri-click strategy appears to lack sufficient novelty for a new contribution. In conclusion, the paper presents an innovative click chemistry-based strategy for developing potential anticancer agents. The approach is scientifically sound, and the focus on oxidative DNA damage is promising. However, expanding on the above-mentioned aspects of specificity and binding

mechanism would significantly strengthen the manuscript. Given these points, the current manuscript seems more suitable for a specialised journal.

A24) We appreciate the reviewer's comments and acknowledge the importance of clarifying both the novelty and mechanistic aspects of this work. As detailed in our responses to Comments Q21 and Q17, we have now included additional analyses and discussion that address the binding mechanism and sequence selectivity in greater depth. We also wish to emphasise that, while the Tri-Click approach was previously applied to nucleic acid-targeting AMNs, the present study constitutes a distinct extension of this chemistry, focusing directly on anticancer applications, single-molecule imaging, and cytological profiling. Collectively, these advances clearly extend the Tri-Click platform beyond prior work and represent a clear progression in the development of copper-based anticancer agents.

Q25) Minor comments:

1. The Figure 4 legends: panels a and c need to be corrected.
2. Information about the presence of multiple-testing corrections is missing from the statistical test details.
3. To improve the readability of the main text, please use the full name for ligand side chains before using the abbreviation.

A25) All three points have now been addressed.